# Application Software That Can Prepare for Disasters Based on Patient-Participatory Evidence: K-DiPS: A Verification Report

**DOI:** 10.3390/ijerph19159694

**Published:** 2022-08-06

**Authors:** Hisao Nakai, Tomoya Itatani, Ryo Horiike

**Affiliations:** 1School of Nursing, Kanazawa Medical University, 1-1 Uchinada, Kahoku 920-0265, Ishikawa, Japan; 2Division of Nursing, Faculty of Health Science, Institute of Medical, Pharmaceutical and Health Science Kanazawa University, Kanazawa 920-0942, Ishikawa, Japan; 3Department of Public Health Nursing, Osaka Medical and Pharmaceutical University, 7-6, Hachonishimachi, Takatsukishi 569-0095, Osaka, Japan

**Keywords:** K-DiPS, smartphone application, web application, disaster preparedness

## Abstract

This paper describes the design and function of an application that enables vulnerable people to provide medical information for use in disasters, and presents the results of an initial test of its usability in Nankoku, Japan. The application consists of two parts: K-DiPS Solo, a smartphone app, and K-DiPS Online, a web application for disaster management by local governments. We asked vulnerable people or their family caregivers to enter medical information into the app on their smartphones and connected this information to a local government application as a demonstration of a disaster response solution that manages information. We targeted a group of 14 healthy older people. The user information that they entered into the app was stored in the cloud via the communication system of the mobile phone. A ledger of vulnerable people for use in the event of a disaster was automatically created on the web application using the information supplied by the individuals. Local government staff corrected the location information, if necessary, by dragging points plotted on a map. This disaster response solution was shown to connect individuals to government offices, and to enable a consistent flow of information from patient details to stocking of supplies, and for simulation, training, and response during disasters.

## 1. Introduction

### 1.1. Frequency and Impacts of Disasters in Japan

Large-scale disasters occur relatively frequently in Japan. The 2016 Kumamoto Earthquake killed 55 people and required more than 180,000 people to be evacuated [1]. The 2018 Hokkaido Earthquake caused a large blackout [2]. This power outage was serious for users of artificial respirators who, in some cases, stopped breathing and had to use a bag valve mask [3]. In recent years, heavy rains have also caused serious damage. For example, rain in Kyushu in 2012 resulted in severe floods and landslides; approximately 400,000 people were evacuated [4] and at least 37 people were killed [5,6]. Disasters caused by abnormal weather linked to climate change may adversely affect the health of vulnerable people, including those with chronic illnesses or specific medical needs. The public hygiene plan highlights the importance of identifying vulnerable groups—people who are ill, pregnant women [7], children, older people, and people living in poverty—in the event of a disaster [8].

### 1.2. Impacts on Medically Vulnerable People and Their Need to Prepare

Medically vulnerable populations generally prepare their medication in case of a disaster, but are unlikely to have a home preparation kit or an emergency evacuation plan to mitigate any issues caused by their health problems [9]. In particular, users of power-hungry medical devices, such as respirators and aspirators, are at risk of dying if a disaster causes a power outage. The 2003 North American power outage put pressure on the resources of emergency departments, because respirator users sought care in emergency hospitals following the failure of their medical devices. Therefore, it is important to forecast the needs of this population and provide effective disaster countermeasures [10]. After the Great East Japan Earthquake of 2011, the majority of pediatric patients using artificial respirators were admitted to medical centers because of prolonged power outages [11]. Local governments therefore need to identify medically vulnerable people, create emergency evacuation plans tailored to their situations, and prepare home preparation kits for them. One of the challenges in providing medical care to vulnerable people affected by Hurricane Katrina was the inability to maintain continuity of medication. This was caused by a lack of information linked to inaccessible medical records and inadequate patient knowledge. This could be addressed by having personal electronic medical records [12].

In recent years, societies around the world have faced growing healthcare challenges linked to aging populations. Fortunately, ICT solutions in healthcare are developing rapidly. There is also an increased use of technology in the medical treatment applied following emergencies and disasters [13]. In Japan, an emergency medical information system (EMIS) is being built with the aim of providing information about appropriate treatments for medically vulnerable populations affected by a disaster [14]. EMIS is an online system for the Ministry of Health, Labour and Welfare, governments in disaster areas, and medical institutions, and is the central information system in the acute phase of a disaster [15]. The system contains information input by doctors, patients, and local governments; this information is stored in the cloud. This provides an “electronic chart function” in the acute phase of a disaster. However, during the Hokkaido Earthquake, and in heavy rain and typhoons, problems related to information sharing, poor operability, few inputs from medical institutions in the disaster area, and a lack of information on people who need power in the event of an outage were encountered. The system therefore needs improvement [16,17]. The Japanese archipelago is situated along the “Ring of Fire,” an area where several tectonic plates meet, and is therefore vulnerable to disasters such as earthquakes, tsunamis, and volcanic eruptions. In recent years, typhoons have regularly hit Japan, often causing heavy rain and floods [18]. It is therefore important for people who live at home but have particular medical needs to be prepared to protect themselves during an emergency, and for local governments to understand the needs of these people so that they can be given the help that they need to stay safe and well.

We have researched and helped to develop a smartphone application that allows these groups to enter their medical information, allowing them to receive assistance from medical personnel in the event of a disaster. This is known as K-DiPS Solo (KDS). We are also developing web applications that connect KDS to the Internet and store information in local government-run clouds. These applications will contribute to disaster preparedness and staff training, and support the confirmation of safety, ongoing treatment, and transportation in the event of a disaster. This app is known as K-DiPS Online (KDO). This paper describes the design and function of KDS and KDO, and presents the results from an initial test of its usability at Nankoku in Kochi, Japan.

## 2. Smartphone Application That Allows People Needing Medical Assistance to Prepare for Disasters

In recent years, several smartphone applications have been developed to help people affected by disasters [19,20,21]. For example, the Federal Emergency Management Agency (FEMA) sends real-time alerts containing weather information and emergency kit checklists. It is also possible to use features such as text messages to find local shelters [22]. The “shelter map” coordinated by the American Red Cross can also be used when searching for shelter [23]. The American Red Cross has a suite of four apps for mobile devices to support disaster preparedness, from how to be safe and prepared in an emergency to how to take care of your pet [24]. “Emergency: Alerts” provides alerts for severe weather, including tornadoes, hurricanes, and floods. “Hurricane” monitors the situation in users’ locations or across the storm track, enabling them to prepare their family and home, find help, and let others know that they are safe. “Tornado” provides a tornado watch and alerts. “Earthquake” provides notifications and alerts in the event of an earthquake or tsunami, enabling people to prepare their families and homes, find help, and let others know that they are safe even during power outages [25]. The “MyHealthDay” mobile app from the Centers for Disease Control and Prevention (CDC) provides up-to-date information on the weather and extreme weather events such as hurricanes and floods [26]. “Notifier Lite” allows someone to search for disaster information that is relevant to his or her location [27]. “GeoNet Quake” provides information and alerts about New Zealand earthquake hazards [28]. These apps help people to prepare emergency kits and provide information to enhance their evacuation behavior. In addition, they provide a variety of alerts for specific hazards. However, current apps are limited to functions that encourage people to prepare for disasters and evacuate. In other words, it is not possible for local government managers to know whether people are taking appropriate disaster prevention actions.

“My Disaster Training” allows people to obtain disaster training information in the event of floods. The aim of this application is for Malaysians to find accurate and up-to-date disaster training information while increasing their knowledge and skills about disaster preparedness before a real disaster occurs [29]. “Edugame” is a game-based application for learning about the disasters that may be caused by earthquakes [30,31]. The “Auckland Civil Defense” app is an educational application for disaster preparedness [32], and an “Android-based mobile learning physics app” has been developed to deepen students’ understanding of tsunamis. Using this app may help in improving students’ problem-solving skills and disaster preparedness [33]. Disaster prevention education using smartphone applications increases students’ and urban residents’ understanding of past earthquakes and tsunamis. People vulnerable to disasters may not be able to attend face-to-face training, but they can acquire knowledge and prepare for disasters through interactive information provision and education using smartphone apps.

Japan has several smartphone applications for sending real-time alerts about weather information, and for helping people to prepare emergency kits in case of a disaster. The Japan Broadcasting Corporation’s “NHK News Disaster Prevention” provides the latest news and weather information on a map, and sends alerts and information about disasters in a live feed [34]. Each municipality has an app that allows residents to receive real-time alerts for weather information and the need to prepare emergency kits. These apps include the Tokyo disaster prevention app [35], Kochi disaster prevention app [36], and Shinsyu disaster prevention app [37]. The disaster prevention information shelter guide allows people to search for the shelter closest to their current location [38]. “Machicare” was developed in the wake of the heavy rains in western Japan in 2018 [39]. This app allows users to understand the risk of a disaster at their place of residence and to manage supplies for three days [40]. These are apps that send weather information and disaster alerts to people, similar to the FEMA and Red Cross apps mentioned above. Machicare aims to improve the provision of supplies for a specific area, but is not based on information about individual people, so there may be a mismatch between the supplies provided and those required in the event of a disaster. Therefore, at present, there is no application that allows people who need regular medical care and help to enter their own medical information and link it to local governments and their disaster management teams.

## 3. Methods

### 3.1. Explanation of K-DiPS

Kanazawa and Kochi Disaster Preparedness System (K-DiPS) is a smartphone application for people who need regular medical care or equipment at home (referred to as vulnerable people). It consists of the KDS smartphone app and KDO web application for disaster management by local governments. Vulnerable people enter their information into the KDS app and send it to the local government’s KDO server, enabling local governments to ensure reliable disaster preparedness based on detailed information about individual patients’ needs [41]. KDS information can be provided directly to medical professionals supporting evacuation, which enables prompt treatment, nursing, and transportation. The app also has a function to enter vital signs and blood test data, making it useful for the daily management of conditions. KDO is an online system that allows local governments to manage disasters by storing the information entered by KDS users in KDO’s cloud storage. That is, municipalities can use KDO to obtain the most up-to-date medical information of local vulnerable people using KDS. This enables municipalities to ensure that they have the right supplies and carry out evacuation training to support preparedness. In the event of a disaster, the app helps to confirm safety, determine rescue priorities, and provide a prompt rescue. The operation of KDS and KDO is illustrated in Figure 1.

### 3.2. Target Area

The study target area was Nankoku city in Kochi, Japan. This is located on the Pacific side of the island and has a population of approximately 46,000. When an earthquake occurs in the Nankai Trough, it may cause a tsunami of 10 m or more in the area on the Pacific coast from Shikoku to Kyushu [42,43]. Figure 2 shows the location of Nankoku and the Nankai Trough (ESRI’s ArcGIS Pro 2.9.3 was used to create Figure 2).

### 3.3. Data Collection

We planned to conduct a demonstration experiment with medically vulnerable people from Nankoku. However, the spread of COVID-19 changed this plan, because medically vulnerable people were at high risk from the disease. Instead, we targeted 14 healthy older members of the Voluntary Disaster Preparedness Organization (VDPO) in Nankoku. Japan’s VDPO was promoted by the Government of Japan to build a more sustainable and resilient community against disasters [44]. The VDPO engages the community in post-disaster drills and other preparatory activities [45]. Many members are retired people. According to a 2016 survey, about 85% of VDPO representatives are over 60 years old [46]. At a VDPO meeting, we explained the purpose and method of the demonstration experiment and recruited participants. These participants were given oral and written explanations and provided written consent.

### 3.4. Explanation of the Demonstration Experiment

#### 3.4.1. KDS Users (Older People and Family Caregivers)

We held a briefing session on how to use KDS for the participants and asked them to undertake a training session. During the demonstration experiment, local government officials and mobile phone company staff built a system to support the use of KDS. We also made it possible for users to contact the principal investigator at any time by mobile phone or email.

To maintain the confidentiality of user information and produce a suitable communication environment for the demonstration experiment, we used the closed NTT DoCoMo network. This closed network was not directly connected to the Internet, and only terminals registered in advance could be connected [47]. We also borrowed dedicated smartphones from NTT DoCoMo that had KDS installed and could be synchronized with KDO.

We distributed these dedicated smartphones (iOS 7 or Android 7) to the participants and asked them to enter their information into KDS. The participants were then asked to tap the sync button on the KDS app. If their information changed during the demonstration experiment, such as if they started taking a new medication, we asked them to correct the information on KDS and tap the sync button each time. The workflow of the system is shown in Figure 3.

#### 3.4.2. KDO Users (Crisis Management Department Staff)

The users’ information was retrieved in the crisis management department via the cloud. The staff of the crisis management department were asked to check the information and ensure that the participants’ addresses were correctly plotted on the map. If the location on the map was not plotted correctly, we asked the staff to correct it using the correction function. During the demonstration experiment period, we asked the staff to confirm the synchronization of user information from KDS to KDO whenever necessary.

After the demonstration experiment was completed, the KDO information was output in CSV format and aggregated. The demonstration experiment lasted from 27 July to 29 October 2020.

## 4. Results

### 4.1. Evaluation of KDS and KDO Design and Function

The user information entered into KDS was passed to the cloud using the communication system of the mobile phone. An administrator password was issued to staff in the crisis management department to enable them to access the protected KDO. The ledger of vulnerable people in the event of a disaster was automatically created on the KDO web application using the information from KDS. This ledger displayed users’ names, dates of birth, genders, and addresses. By clicking on the information list, staff could access user photos, emergency contacts, supporter details, information about support needed with activities of daily living (ADL), medical information, and notes. The users’ addresses were geocoded and plotted on the KDO map. The location information was corrected, if necessary, by dragging the points plotted on the map.

Figure 4 and Figure 5 present the inputs to the KDS app and the corresponding KDO data. KDS screenshots are shown in Figure 6 and Figure 7. There is no English version of KDS and KDO, so the Japanese version is shown.

### 4.2. Aggregation of Information in KDO

Overall, 14 people participated in the demonstration experiment. The average age (standard deviation) was 71.1 years (10.8). Six people were male (42.9%), three were female (21.4%), and five did not specify a gender (35.7%). The users’ primary contact was their wife (four people), father, mother, eldest daughter, child, and niece (one person each); four participants did not specify a primary contact. Six people lived with their primary contact. A main disease was given by 10 people. One participant had no main illness and was not on medication. Nine people regularly went to the hospital to receive dosing treatment. Four had hypertension and two had hyperlipidemia. Gout, mild dementia, chronic renal failure, interstitial pneumonia, pulmonary suppuration, diabetes, and atrioventricular block were each reported by one person. (Note that multiple answers were possible.)

## 5. Discussion

The study participants were all able to enter information into KDS on their smartphones. This process would enable them to quickly inform the disaster medical assistance team (DMAT) of their medical condition in the event of a disaster. Nara Prefecture in Japan recommends that children in need of medical care prepare and carry a written list of necessary supplies so that they can pass this to their supporters in the event of a disaster [48]. However, a paper-based method of passing information carries the risk of (1) requiring a written amendment each time the information is updated, which is burdensome, and (2) forgetting to carry the document. Using KDS via a smartphone overcomes these two issues, but faces the risk that the phone may run out of power. Disaster countermeasures for vulnerable people include the identification of high-risk clients and the provision of written materials and recommendations [49]. Measures are therefore being taken to create a written emergency kit [50]. However, this is likely to be most useful for people whose physical condition is stable, which usually means they are healthy. Written emergency kits are not appropriate for home-based patients who require advanced medical services because of their frequent changes in physical condition. In an emergency situation during a disaster, it is important for vulnerable people to be able to accurately inform DMAT of their medical condition. Notification of such information using KDS may also help people with respiratory syndromes following the COVID-19 pandemic to be treated promptly and appropriately. However, older people who are less familiar with or have little access to technology need to be encouraged and assisted to enter and update information. Thus, it is recommended that tasks such as information input, confirmation of information updates, and support be added to the existing workload of visiting nurses, case managers, and public health nurses. As a result, groups that are unfamiliar with technology can be identified and their support enhanced. The rapid availability of patient information with KDS is beneficial to patients, doctors, and healthcare professionals. Proper communication from vulnerable people can contribute to better medical decisions.

The link between KDS and KDO may be useful for disaster preparedness, prompt treatment, and the provision of nursing and transportation in an emergency. The stock of supplies for disasters in Japan is calculated by each local government using information from previous disasters, knowledge of the population, and damage estimation. Kochi Prefecture, which includes the target area of this study, uses this calculation method [51]. Because this approach does not draw on actual information about people who need medical support at home, there may be a mismatch between the supplies stored by the local government and those required by vulnerable people in an emergency. By connecting KDS to the system, vulnerable people’s needs can be shared with the crisis management department of the local government. This solution gives disaster response managers detailed information about vulnerable people. After Hurricane Katrina, disaster response management decisions were hindered by a lack of coordination, inadequate information flow, and other obstacles [52]. The disaster response solution offered by KDS and KDO provides a consistent flow of information from vulnerable people to local governments, supporting better management of supplies and enabling simulations, training, and appropriate responses in the event of a disaster. Expanding the use of this solution to elderly people, pregnant women, and other vulnerable residents will make it possible to stock municipal supplies based on accurate information about local residents’ needs. This will help to reduce costs by avoiding unnecessary stockpiling.

Analyzing the data added to KDS in this demonstration experiment, there were some items that were unused. The target of the demonstration experiment was older people, and a dedicated terminal was used. It is possible that particular items were not input because people were unfamiliar with that type of smartphone. If home-based patients start using KDS on their smartphones, they may need some support. Above all, assistance is needed to enter medical information. This may require someone who is familiar with KDS, such as a visiting nurse. Such support would ensure that nothing was missed, because visiting nurses would be familiar with the medical and home needs of vulnerable people. Lack of incentives for information sharing between individuals and organizations may be a barrier to information sharing for disaster preparedness [53]. It is therefore advisable to provide an incentive for visiting nurses to help vulnerable people prepare for a disaster. By making the provision of support routine, it may be possible to reduce the anxiety of home-based patients in the event of a disaster, and contribute to appropriate disaster countermeasures. Geocoding the participants’ addresses was also a challenge in terms of accuracy. The addresses could be corrected manually, but further corrections and checks would be needed to help confirm the safety and evacuation plan in the event of a disaster.

This study had several limitations. The number of people targeted for the demonstration experiment was small (14), making the experiment insufficient for verifying the applicability or usability of the app. Therefore, our results cannot be generalized. Second, the target group was limited to healthy older people living in the community, none of whom needed medical care. Hence, it was only possible to verify certain items of medical information in KDS. Third, this experiment used a dedicated smartphone for the experiment and a closed network line, which is different from the communication environment that would be used in reality. In the future, we will carry out demonstration experiments using standard communication networks. Finally, the spread of COVID-19 meant that group interviews on user experience and usability were canceled. To establish a strong reputation for this app, a further demonstration experiment with a significant increase in the number of participants is required. However, despite these restrictions, this study has shown that the app may help vulnerable people prepare for disasters. It may also help municipalities to identify home-based patients in the area and ensure that they remain safe and well in the event of a disaster.

## 6. Conclusions

Municipalities could encourage the use of KDS among home-based patients and other vulnerable people as a means of managing the supplies and support required by these groups in the event of a disaster. Increasing the number of home-based patients who enter their medical information in KDS and send it to KDO will improve the disaster prevention capability of the area by improving the evidence base for planning and disaster preparedness.

## Figures and Tables

**Figure 1 ijerph-19-09694-f001:**
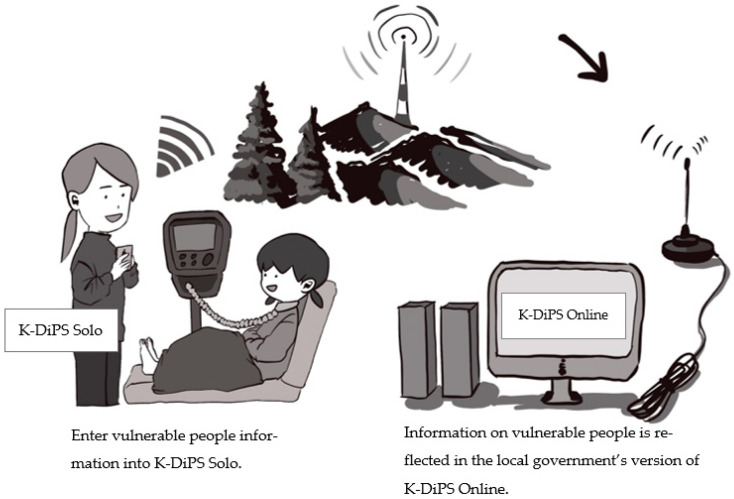
Operational process of KDS and KDO.

**Figure 2 ijerph-19-09694-f002:**
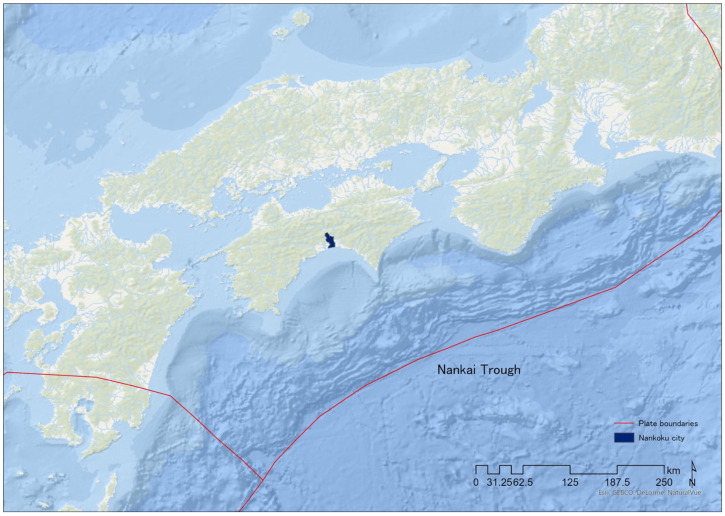
Target area of K-DiPS demonstration experiment in Japan.

**Figure 3 ijerph-19-09694-f003:**
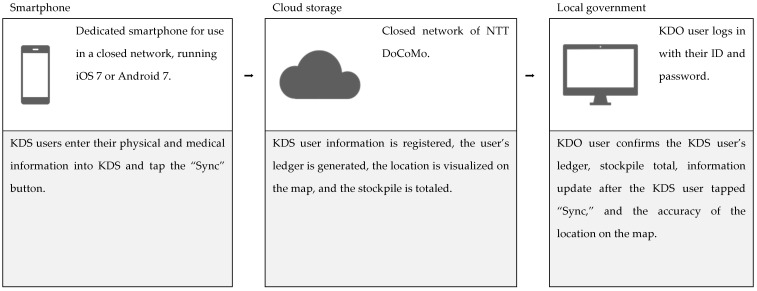
Workflow of the system.

**Figure 4 ijerph-19-09694-f004:**
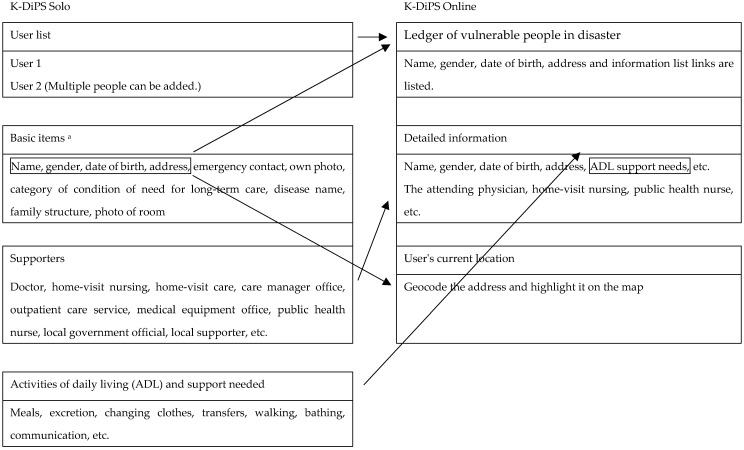
Basic information included in KDS and corresponding data in KDO. ^a^ The app contains a function to allow users to take a picture and save information.

**Figure 5 ijerph-19-09694-f005:**
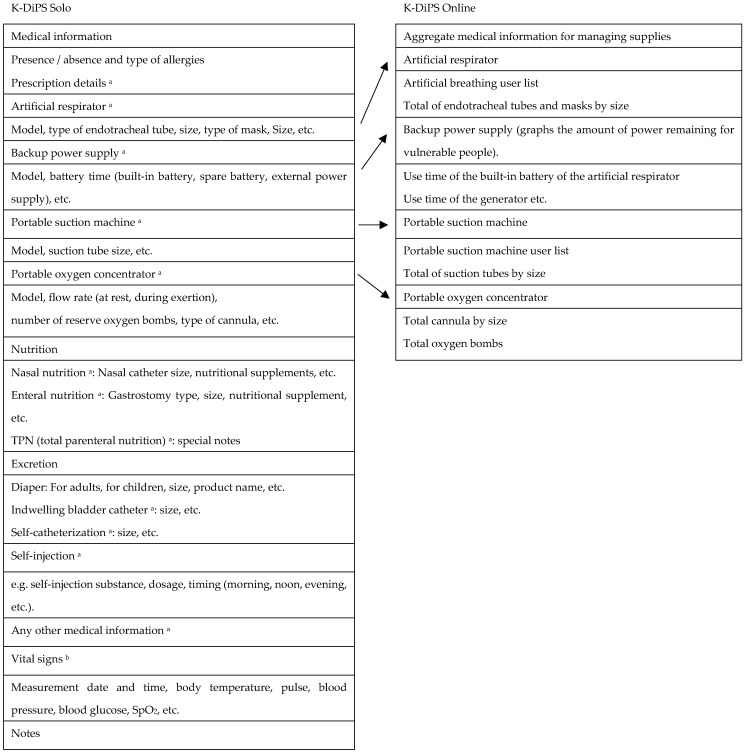
Medical information included in KDS and corresponding data in KDO. ^a^ The app contains a function to allow users to take a picture and save information. ^b^ The function that can record vital signs was implemented in April 2022 with the version upgrade of KDS, and cannot yet synchronize with KDO.

**Figure 6 ijerph-19-09694-f006:**
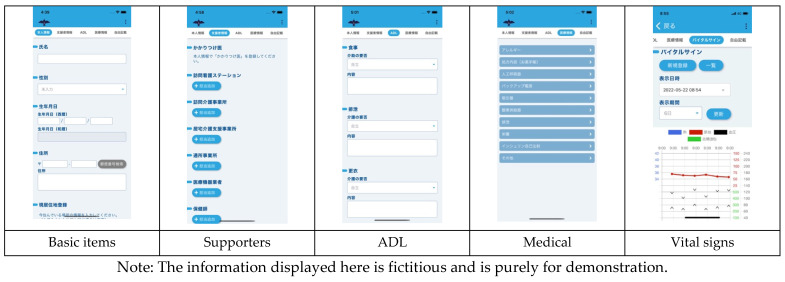
Screenshots of “Basic items,” “Supporters,” “ADL,” “Medical,” and “Vital signs” in KDS.

**Figure 7 ijerph-19-09694-f007:**
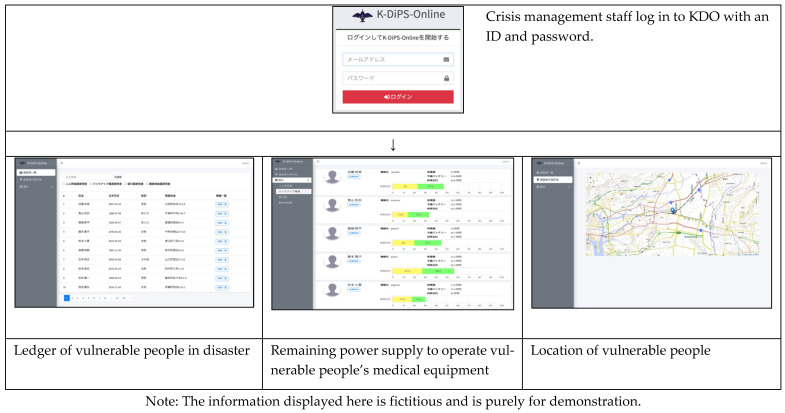
Screenshots of “Ledger of vulnerable people in disaster,” “Remaining amount of backup power supply,” and “Display of users’ whereabouts” in KDO.

## Data Availability

The data analyzed during this study are included in this published article. Further inquiries can be directed to the corresponding authors.

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
