# Peer review of "Application Software That Can Prepare for Disasters Based on Patient-Participatory Evidence: K-DiPS: A Verification Report"

_ijerph, 2022, doi:10.3390/ijerph19159694_

Round 1
Reviewer 1 Report
Summary:
The authors introduce an app-based emergency medical contact system in a local town in Japan, showing its usability with 14 elderly people recruited for the demonstration experiment.
General Comments:
A literature review of how experts, users, and the media’s opinion on the apps is warranted. What do you learn from those disaster prevention, notification, and education apps? I think beyond an introduction of other apps, you need to review, synthesize and comment on their functionalities. This would serve as basis for you to evaluate the appropriability of your very own app.
In the wake of COVID-19 pandemic, I think you should discuss how your app would be attuned to cater for the needs of those who suffer from respiratory syndromes and need medical attention.
Nowadays people use smartwatches to measure heartbeat rate, body mass index, and blood oxygen level, etc. Does your app connect to those devices to allow for instantaneous measure of body conditions, synchronized to the cloud to enable medical personnel to evaluate on-time?
Since user information is stored on cloud servers, you need to evaluate security of the cloud service providers to ensure that sensitive information will not be breached and compromised.
Typically, 14 observations are far from statistically sound to draw meaningful conclusions. I am looking for 120 participants, to say the least. Please cite the literature to justify your sample size (rather than just saying that this is one of your limitations).
Section 4.2 needs to be more elaborated about the medical conditions to facilitate an unbiased evaluation. For example, do you self-select the participants to have a “diverse” background? Put it simply, can you mention how the 14 participants are randomly chosen to avoid selection bias? By the way, may I have a look at your letter of consent?
In addition, did any one of the 14 participants run into situations that need immediate medical attention during the observation period? If no, then your app has not really been put into test.
Specific Comments:
In line 79, you mention “we are developing”. So, are you the first-hand developers of the system or authors who write about engineers who develop it? Please clarify to avoid any conflict of interest.
In line 109-10, you mention “help people manage (themselves)”, and I assume an objective there. Also, can you explicitly call out the names of the applications in lines 111-12?
In line 219 and 223, you have the single purpose of demonstration, so no plural form “purposes”.
In line 305, “help” is used as an intransitive verb, so paraphrase as “to help them manage …”
Reviewer 2 Report
In this paper the authors presented a work on app (mobile and web) design and use for elderly/vulnerable people. The app is used by users to provide medical information to local government websites which is then used for disaster response. The data stored in the local government sites can then be updated by the governmental staff to increase the accuracy of data.
The motivation of this research is presented clearly in the introduction section. The tendency of natural disasters in many countries including the country from where the authors come from demonstrates a strong need for such applications for disaster management. The related work also highlights similar work/app that are quite numerous. Their main characteristics are also presented to show the uniqueness of this proposed work.
The Methods section is the main contribution of this research. The research methodology is explained alongside the data collection, the users description and the target area. K-DiPS and its online version KDO are also briefly explained here for the background.
The experiment and result section demonstrates the workflow functionalities of the system. Some interfaces are also presented to show how data is collected by the users and presented on the apps. Finally qualitative discussions is presented.
Overall the paper is quite interesting, but one thing missing is a solid evaluation. Having a system that works is a good start, but evaluation should be beyond this. Surveys can be conducted to the users to validate the user experience. Some functional testing is also a simple was of evaluation from the field of software engineering.
Round 2
Reviewer 1 Report
Line 120. Can you clarify what you mean by "mobile learning physics app"?
